# Wearable Hand Module and Real-Time Tracking Algorithms for Measuring Finger Joint Angles of Different Hand Sizes with High Accuracy Using FBG Strain Sensor

**DOI:** 10.3390/s20071921

**Published:** 2020-03-30

**Authors:** Jun Sik Kim, Byung Kook Kim, Minsu Jang, Kyumin Kang, Dae Eun Kim, Byeong-Kwon Ju, Jinseok Kim

**Affiliations:** 1Center for Bionics, Korea Institute of Science and Technology, Seoul 02792, Korea; kjs414@kist.re.kr (J.S.K.); cfg10x6p1@kist.re.kr (B.K.K.); minsujang@kist.re.kr (M.J.); kyuminkang@kist.re.kr (K.K.); 2Display and Nanosystem Laboratory, School of Electrical Engineering, Korea University, Seoul 02841, Korea; 3School of Mechanical Engineering, Yonsei University, Seoul 03722, Korea; 4School of Chemical Engineering, Sungkyunkwan University, Suwon 16419, Korea; 5Department of Electrical Engineering, Korea University, Seoul 02841, Korea

**Keywords:** fiber Bragg grating strain sensor, algorithm, hand motion capture, real-time tracking

## Abstract

This paper presents a wearable hand module which was made of five fiber Bragg grating (FBG) strain sensor and algorithms to achieve high accuracy even when worn on different hand sizes of users. For real-time calculation with high accuracy, FBG strain sensors move continuously according to the size of the hand and the bending of the joint. Representatively, four algorithms were proposed; point strain (PTS), area summation (AREA), proportional summation (PS), and PS/interference (PS/I or PS/I_α). For more accurate and efficient assessments, 3D printed hand replica with different finger sizes was adopted and quantitative evaluations were performed for index~little fingers (77 to 117 mm) and thumb (68~78 mm). For index~little fingers, the optimized algorithms were PS and PS/I_α. For thumb, the optimized algorithms were PS/I_α and AREA. The average error angle of the wearable hand module was observed to be 0.47 ± 2.51° and mean absolute error (MAE) was achieved at 1.63 ± 1.97°. These results showed that more accurate hand modules than other glove modules applied to different hand sizes can be manufactured using FBG strain sensors which move continuously and algorithms for tracking this movable FBG sensors.

## 1. Introduction

Accurate measurement of hand motions is useful for various applications such as rehabilitation, virtual reality (VR) and augmented reality (AR) technology, and robotics [1,2,3,4]. For example, after a stroke, the damaged functions of the brain can be improved through continuous and repetitive finger rehabilitation [5,6]. The hand serves as a conduit for direct interaction between objects in a VR/AR world and users in the real world [7]. For robot-assisted surgery (RAS), accurate measurement of the finger movements is needed to ensure smooth teleoperation with the surgical robot and prevent mistakes [8]. Thus, studies have focused on using various sensors and equipment such as optical marking methods (i.e., camera recognition), inertial measurement unit (IMU) sensors, electrical resistance strain sensors, and fiber optic sensors to accurately measure and evaluate finger movements [1,2,3,9,10].

The optical marking method visually identifies changes in marked positions using an optical camera. This method can be used with marker recognition algorithms to check the movement of joints [2]. Although it provides excellent accuracy, the installation cost of equipment such as cameras is too high, and it is best suited to limited spaces where user motion is limited [11]. The algorithms are also very complex [12,13]. Research on overcoming the limitations of the optical marking method has focused on fabricating glove-type sensor modules where sensors (e.g., IMU, strain sensor, optical fiber sensor) are attached directly to the glove to measure the movement of finger joints. Using gloves ensures high accuracy while increasing the user’s freedom of movement.

The IMU sensor has multiple degrees of freedom and measures linear acceleration and angular velocity along three axes [14,15,16]. For a more accurate measurement, an inertial navigation system (INS) also uses triaxial magnetic sensors and barometric pressure sensors [16]. While an IMU sensor can measure finger movements at a relatively low cost and is easy to install, the measurement data can be affected by electromagnetic fields, which can result in large errors. Errors from magnetic or angular velocity sensors accumulate over time to cause drift [17,18]. This is a chronic drawback of IMU sensors that many researchers are working on to reduce; however, it cannot be eliminated because it is an accumulation of errors over time [17]. Another disadvantage of IMU sensors is the complexity of the calculation process to locate an object [19]. The *x*, *y*, and *z* positions and the yaw, pitch, and roll angles are obtained by applying compensation and the Kalman filter to acceleration and angular velocity sensor data; however, the human hand has 27 degrees of freedom, and some joints depend on the movement of other joints [20]. Obtaining the position/rotation angle information for each finger phalange becomes very complex if all finger movements are acquired with multiple IMU sensors and data are fused and decoupled during the calculation process. An alternative approach is to attach a resistive strain sensor to the position of each finger joint to directly measure the strain of a bending joint [21,22,23]. While this approach reduces drift and has a light calculation load, the positions of the strain sensors on the glove are fixed; thus, it is difficult to produce a hand module with excellent accuracy for users with different hand sizes.

Unlike previous studies that focused on accurately measuring finger joint angles, recent studies have focused on developing a dexterous hand module with a small and repeatable measurement error for various users [6,22,24]. Wise et al. used optical fibers to fabricate finger angle tracking gloves with three different sizes so that users can select the right size themselves [25]. Borghetti et al. used a slightly different approach [24] and developed a calibration procedure to measure the joint angles based on the hand size of the new user. However, these studies did not assess the angle error when various users equip the glove. Even if different types of gloves are made, people have a wide range of hand sizes. Thus, a system needs to be developed that can use the same calibration process to measure any hand size. Li et al. developed a hand module that uses a stretchable bending sensor and achieved an absolute angle error of 4.5–8.0° for a hand size range of 17–21 cm [22]. Although they used stretchable sensors, the sensor positions in the hand module were still fixed, so reducing the measurement angle error would be difficult when applied to various hand sizes. Jha et al. used a fiber Bragg grating (FBG) sensor to measure the angles of two different hand sizes; they achieved a very small angle error of 0.13° with a mechanical setup and 0.67° with a human hand with very small maximum standard deviations of 0.30 and 0.67°, respectively [6]. However, they did not measure the joint angles of the thumb and little finger in real time. In addition, they used the angle of the distal interphalangeal (DIP) joint to approximate the angle of the proximal interphalangeal (PIP) joint. Thus, more research is needed to realize an extremely small angle error that can be used for practical applications in the industry.

As indicated above, hand motion tracking glove modules with high accuracy need to be developed for various hand sizes. Such equipment needs to be applicable to different rehabilitation patients, VR/AR users, and surgeons [4,5,8]. In particular, RAS requires high accuracy. This study focused on evaluating the performance of a wearable hand module. Unfortunately, the scope of this study was limited to adult males. A key reason for this is the length of the silicone ring structure. If the silicone ring structure is reduced to match the size of an adult female’s hand, this causes the silicone to reach its maximum tension before a male can clench his fist. Meanwhile, if the silicone ring structure is enlarged to match the size of an adult male’s hand, the thimble can easily escape from the end of the finger when a female opens her fist. Thus, 11 adult males aged between 26 and 48 years old were evaluated to determine the range of the hand size, and the angle error of the wearable hand module was measured.

Configuring a versatile system that is applicable to various hand sizes and has a small angle error is difficult. The first problem is the sensor attachment in the hand module [26,27,28]. Since the sensor positions in the hand module do not change, the relative positions of the joints and sensors differ among users with different hand sizes, which does not allow repeatable data to be obtained. The second reason is the algorithm. Even if the attachment problem is resolved, the sensor data are measured differently for different users. Algorithms need to be developed that can identify trends in data despite differences in the measurement among users with different hand sizes.

In this study, a system was developed for accurate and real-time tracking of the finger joint angles of adult males regardless of hand size. Hand replicas were 3D printed for different finger sizes (index-little fingers: 77–117 mm, thumb: 68–78 mm) because using actual hands could introduce many critical variables such as hand tremors, temperature, and limitations of the goniometer or other sensors. Unlike the traditional method of fixing the sensor position in the glove, a smaller angle error was achieved by developing an algorithm that converts data from movable sensors for real-time calculation. According to the literature, the allowable angle error of the finger joint for clinical purposes is less than 5° [29]. Therefore, the objective of this study was to realize a maximum angle error of no more than 5°. In the experimental results, the maximum angle error of the wearable hand module with the movable FBG strain sensor was 4.6°; the mean angle error and standard deviation were 0.47 ± 2.51°, and the mean absolute error (MAE) was 1.63 ± 1.97°.

## 2. FBG Strain Sensor and Hardware Design

### 2.1. Principle of the FBG Sensor

The FBG sensor is a thin optic fiber strand consisting of core, cladding, and buffer layers with lattice patterns evenly engraved on the core [30]. When the core emits a broadband light, only a certain wavelength is reflected from the engraved lattice pattern. This reflected wavelength is called the Bragg wavelength, and it expresses the relationship between the effective refractive index of the fiber core and distance from one lattice to the next lattice [31]:(1)$$\lambda_{B}=2n_{eff}\Lambda$$

Here, $\lambda_{B}$ is the Bragg wavelength reflected by the lattice pattern, $n_{eff}$ is the effective refractive index of the fiber core, and Λ is the period of the lattice pattern. Equation (1) shows that the Bragg wavelength is linearly proportional to the effective refractive index of the fiber core and distance between lattice patterns. The effective refractive index of the fiber core is a constant value determined at the time of manufacture. The Bragg wavelength changes with the distance between grid patterns, which is affected by temperature or strain. Therefore, an FBG sensor can be used to detect strain or temperature. In this experiment, a quantitative assessment was performed with a hand replica at the same temperature as the environment, not with an actual human hand. In addition, because the data were measured after the temperature was stabilized, the effect of the temperature was not considered to be significant. The variation in the Bragg wavelength due to strain can be expressed by [32]:(2)$$\Delta \lambda_{B}=\lambda_{B}\left(1-\rho_{\alpha }\right)\Delta \varepsilon_{axial}$$
where $\lambda_{B}$ is the shift in the Bragg wavelength due to axial strain, $\rho_{\alpha }$ is the coefficient of photo-elasticity of the optical fiber, and $\Delta \varepsilon_{axial}$ is the axial strain. Thus, the change in the Bragg wavelength is only affected by the axial strain.

### 2.2. Design of the Embedded FBG Strain Sensor

To measure the angle of a finger joint, an FBG sensor should be able to measure not only the axial strain but also the bending strain. Researchers have studied attaching an FBG sensor to a surface [33,34] or embedding it in a polymer matrix [35,36] to measure the bending strain. An adhesive [33] or polyimide backing patch [37,38] is usually used to attach an FBG sensor to a bending surface. In this case, nonuniform hardening of the adhesive may cause nonuniform bending at the FBG node or reduce the durability of the sensor over a long bending time because of the reduced adhesion of the tape [33,34,37]. Therefore, FBG strain sensors are embedded into a polymer matrix to measure the bending strain more reliably [39,40]. When an FBG strain sensor is bent, the strain delivered to the FBG nodes increases by the offset distance between the sensor and neutral bending axis. He et al. determined the relationship between the position of the FBG sensor from the bending center and the change in the Bragg wavelength if the sensor is subjected to a bending strain [39,40]:(3)$$\Delta \lambda_{B}=\lambda_{B}\left(1-\rho_{\alpha }\right)C\cdot h_{t}$$
(4)$$x=h_{1}-\left(h_{t}/2\right)=h_{1}-\left(h_{1}+h_{2}\right)/2$$
where C is the bending curvature. $h_{1}$ and $h_{2}$ are the positions of the optical fiber core from the top and bottom, respectively, of the polymer matrix and are constant values determined during the manufacturing process. $h_{t}$ is the summation of $h_{1}$ and $h_{2}$, and refers to the thickness of the FBG strain sensors. The change in the Bragg wavelength of the embedded FBG sensor is predominantly affected by the curvature C and varies from the neutral bending axis to the offset distance x of the FBG. As you can see, the data of FBG strain sensors can be changed by either the curvature C or the offset distance x. The curvature C is not a value we can designate, but an external factor that we should measure with sensors. So only what we can handle is the offset distance x. By (3) and (4), as offset distance x increases, peak wavelength change increases. This means that the sensitivity of the FBG strain sensor increases. Moreover, too much sensitivity of the FBG strain sensor can cause a collision between peak wavelengths of adjacent nodes, and if the sensitivity is too low, the effect of noise will increase. Based on this principle, Figure 1 illustrates the concept of measuring the angles of finger joints.

In this study, considering the abovementioned issues regarding selection of appropriate offset distance, the core of the FBG strain sensor was placed at an offset distance x of approximately 53 µm from the neutral axis within a rectangular epoxy matrix with a height of approximately 250 µm (Epotek-301, Epoxy Technologies Inc., Billerica, MA, USA). The sensor was rectangular in shape with a width of 2 mm, and it measured the bending strain only in the upward and downward directions, not the leftward and rightward directions. The embedded FBG sensor (FBGS Technologies GmbH, Jena, Germany) had FBG nodes with a 3 mm length engraved on a 125 µm diameter optical fiber coated with Ormocer ^®^. The distance between FBG nodes was 19 mm. Due to differences in finger lengths, nine nodes of the FBG sensor were used for the index, middle, and ring fingers; seven nodes were used for the little finger; and five nodes were used for the thumb. Table 1 details the specifications of the FBG sensors.

### 2.3. Fabrication of the Wearable Hand Module

Figure 2 shows the detailed design and configuration for a wearable hand module manufactured with FBG strain sensors. First, to protect against ambient temperature and impact, the FBG strain sensors are fixed inside the thermoplastic polyurethane (TPU) guide (Cubicon 3DP-310F, Cubicon Inc., Gyeonggi-do, Korea). While the TPU guide is fixed at the nail part, the FBG sensor is attached to the opposite end inside of the TPU guide. If the FBG sensor is attached to the nail part where the TPU guide is also attached, an axial strain will be applied to the FBG strain sensor when the finger bends. To avoid this problem, the FBG strain sensor is attached to the opposite end of the fixing points of the TPU guide. Using this method, only the strain caused by finger bending is transferred to the FBG strain sensor. The TPU guide is designed to be movable when the hand is clenched into a fist or the palm is opened with a passage structure (flattener) made of VeroWhite Plus (Objet350 Connex, Stratasys, Eden Prairie, MN, USA) on the back of the hand. Teflon films minimize friction at the contact surface with the TPU inside the flattener. The flattener flattens the FBG nodes on the back of the hand, which helps distinguish the curvature of the metacarpophalangeal (MCP) joint more effectively. The fingertips are connected to the flattener structure with silicone rings, which hold the FBG strain sensors within a certain distance from the fingers. This mitigates the friction inside the flattener because the resilience of the silicone ring structure is transferred to the TPU when the fist is clenched. The wearable hand module was used to quantitatively evaluate a hand replica with different finger sizes. A rendering program was used to track the joint angles in real time as the module was worn by different hands.

### 2.4. Evaluation of the FBG Strain Sensor

Bending tests were performed to assess the linearity and durability of the FBG strain sensor for measuring the joint angles of the fingers. An experiment was carried out with a bending tester that could apply an angle of 90°, and the sensor was bent 10,000 times using a stepper motor. Not many nodes corresponded to the curvature of the bending tester, so 10,000 bending cycles were performed on two nodes, and the linearity was tested before and after the experiment. The Bragg wavelength during the 10,000 bending cycles was recorded to observe any significant changes in the Bragg wavelength. Figure 3a shows the linearity of the sensor data before and after the high-cycle bending test of the FBG strain sensor. The FBG strain sensor indicated no major problems with linearity before and after the 10,000 cycle test with $R^{2}$ values of 0.997–0.998. Figure 3b shows no significant change in the Bragg wavelength during the bending test. However, if the sensor data are obtained as a function of strain and the sensor is attached to the wearable hand module, the linearity of the system results may be significantly degraded. This is because the curvature acting on the node must be constant; however, the curvatures of the finger joints are very close to each other and can have different values. This problem cannot be solved solely through the sensor design and needs to be addressed by using an algorithm to process the raw sensor data. Section 3 presents the process of reading data from the FBG strain sensor and using algorithms for calculations with the data.

## 3. FBG Interrogation System and Real-Time Tracking Algorithm

### 3.1. Interrogation Process

A commercial interrogator (Smart Fibres Limited, Bracknell, UK) was used to measure the changes in the Bragg wavelength of the C and L bands for the FBG strain sensor in four channels. As discussed in Section 2, bending the fingers changed the Bragg wavelength of the FBG node within the FBG strain sensor. The interrogator was used to check the shift in the Bragg wavelength of the FBG nodes and perform the algorithm calculations in real time. Figure 4 shows the process of converting the shift in the Bragg wavelength of the FBG strain sensors within the wearable hand module with the commercialized interrogator.

To calculate the algorithm, Bragg wavelength data are sent from the smart fiber interrogator to a PC via User Datagram Protocol (UDP) communication. A tracking algorithm in the Microsoft Foundation Class (MFC)-based program translates the Bragg wavelength data into the angle for each joint of the finger. The algorithm calculates the change in the Bragg wavelength of the FBG nodes attached to the wearable hand module every 10 ms within the thread function. The buttons in the dialog box are used to select one algorithm and track the finger joint within 20–40 ms.

Before the algorithm can be performed, the reference data for the finger joint angles when the user’s fist is clenched need to be entered. The required reference data are the angles of the MCP, PIP, and DIP joints for a clenched fist and the Bragg wavelengths for a clenched fist and open palm. These five input variables are used to determine the exact finger joint angles between a clenched fist and open palm in real time. If the peak shift is outside this range, it is not used as data. For certain nodes, very small peak shifts occur between the clenched fist and open palm, and this data can lead to very large angle measurements. Thus, they are regarded as noise and excluded. If the denominator is very small, the angle will be infinite for an open palm. In this case (<1°), the angle is input into the 3D hand graphic using an alternative algorithm (e.g., PTS, AREA, PS) that does not have a denominator of zero for any given angle.

### 3.2. Proposed Algorithm

Various algorithms have been studied to accurately track the angles of finger joints. However, universally applicable algorithms are difficult to implement because of differences in the type of hand module, sensors measuring strain or curvature, and reference calibration method. Park et al. fabricated a hand module from Eco-flex and used changes in the resistance of the internal conductive liquid metal to convert the strain from a bending finger into the angle of a finger joint [7]. The key concept of their algorithm was decoupling to distinguish the flexion/extension motion from the abduction/adduction motion. Wang et al. used 3D printing to fabricate structures that they attached to the fingers to track the angles of finger joints during abduction/adduction, circulation, and flexion/extension motions [9]. They used the weighted average method to develop an algorithm for calculating angular data by comparing the voltage amplitudes obtained from two or more channels. However, the algorithms used in these studies were data isolation processes to obtain pure flexion or pure abduction; in contrast, the aim of the present study was to accurately measure the flexion and extension of each joint. Park et al. tried to distinguish the flexion/extension motion of a joint by using a flexible wire and a linear potentiometer [41]. The resistance of the linear potentiometer attached to the PIP joint during a clenched fist was calculated by subtracting the angle of each joint, while the resistance of the potentiometer attached to the MCP joint was added. Thus, they mainly calculated the joint angle through a sensor/spring system where the spring structure was directly attached to the sensor and the strain in the axial direction was measured [6,41]. In the present study, a spring structure was similarly used within the wearable hand module; however, the spring does not pull the sensor but instead helps the sensor easily enter the flattener structure when the motion shifted from flexion to extension. Since the FBG strain sensor moves continuously inside the flattener structure according to the finger’s movement, the positions of the FBG nodes measuring each joint (especially the MCP joint) can move. Experiments showed that the FBG node positions for the DIP and PIP joints did not significantly differ with respect to the finger length and motion because they were close to the fixed point. However, the FBG node positions for the MCP and interphalangeal (IP) joints varied greatly. Thus, the key concept of the proposed algorithm is to analyze the change in sensor data due to changes in these positions. Figure 5 details the positional change process.

In contrast to the traditional method of directly measuring the bending strain of a joint by attaching a strain sensor, the periodically engraved FBG nodes change their position as the DIP, PIP, and MCP joints bend (especially the PIP joint). For the DIP and PIP joints, the FBG node positions do not change much between a clenched fist or open palm, but the node positions applicable to the MCP joint change significantly. Therefore, using only one FBG node for the MCP joint introduces too large an error. If an FBG node is fixed to each joint, then the joint angle can be converted to data that can easily be obtained from a single node. However, if it is not fixed and is moving, the nodes corresponding to each joint (especially the MCP joint) will also change, and an algorithm is required to convert the corresponding FBG node positions into the angle in real time. Various algorithms were considered for this purpose, but the most basic point strain methods are introduced first for comparison.

The most basic algorithm for tracking the joint angle is the point strain (PTS) method, which uses data from only one FBG node for each joint. The peak shift that occurs at one node for each joint between a clenched fist and open palm is stored; when the current peak shift (current peak shift = (current Bragg wavelength)—(Bragg wavelength during an open palm)) increases linearly, the measured joint angle increases linearly from the open palm to a clenched fist. This is expressed by:(5)$$\theta_{mea}=\frac{\Delta \lambda_{i\_c}}{\Delta \lambda_{i\_m}}\times \theta_{r-p}+\theta_{p}$$
where $\theta_{mea}$ is the joint angle calculated in real time by the sensor/algorithm. $\theta_{r-p}$ and $\theta_{p}$ are the actual angles (angle measured by the goniometer for a clenched fist that should subtract the initial angle for an open palm $\theta_{r-p}$) for the clenched fist and open palm (initial angle $\theta_{p}$). For an actual human hand, the initial angle ($\theta_{p}$) with an open palm can be set to 0 because of the difficulty in measuring the exact initial joint angles. For the hand replica, however, the initial angle measured in the quantitative evaluation was entered. $\Delta \lambda_{i\_c}$ is the current peak shift of the *i*th node. $\Delta \lambda_{i\_m}$ is the maximum peak shift between the open palm and clenched fist of the *i*th node. Thus, $\Delta \lambda_{i\_c}/\Delta \lambda_{i\_m}$ has a value from 0 to 1 and is called the bending percentage (B%) of the *i*th node.

In the aforementioned method, it is necessary to assume that only one FBG node corresponds to each joint when moving from an open palm to clenched fist. However, because two or more FBG nodes can be matched to each joint and the FBG node positions change continuously as the fingers bend, matching only one node to each joint could be a fatal error. Therefore, setting two or more FBG nodes to measure each joint angle can help avoid this error. This is the concept behind the area summation method (AREA):(6)$$\theta_{mea}=\left(\frac{\Delta \lambda_{i\_c}}{\Delta \lambda_{i\_m}}+\ldots +\frac{\Delta \lambda_{n\_c}}{\Delta \lambda_{n\_m}}\right)\times \frac{1}{\left(n-i\right)+1}\times \theta_{r-p}+\theta_{p}$$

In contrast to the PTS algorithm given in Equation (5), AREA requires two or more nodes for each joint angle. Thus, the number of B% (bending percentage) terms increases ($\Delta \lambda_{i\_c}/\Delta \lambda_{i\_m}$, $\Delta \lambda_{n\_c}/\Delta \lambda_{n\_m}$). The *i*th node is the first node that corresponds to a joint, and the *n*th node is the last node to correspond to the joint. For example, nodes 1 and 2 (i=1, n=2) may correspond to the DIP joint, nodes 3 and 4 (i=3, n=4) may correspond to the PIP joint, and nodes 5, 6, 7, and 8 (i=5, n=8) may correspond to the MCP joint for the index-little fingers. Equation (6) separately adds B% of the i and n nodes ($\Delta \lambda_{i\_c}/\Delta \lambda_{i\_m}$, $\Delta \lambda_{n\_c}/\Delta \lambda_{n\_m}$) and distributes the number of nodes at the reference angle ($1/\left(n-i+1\right)\times \theta_{r-p}$), which does not greatly differ from simple summation. Equation (6) can greatly reduce errors that may occur with Equation (5). AREA adds the changes at each node (*i* – *n*) equally, so it is recommended if all FBG nodes are well-seated on the curvature of the joints.

However, depending on the size of the hand or bending, the FBG nodes may be dislodged from a joint. If the FBG nodes are not well-seated and dislodged from the joints, they should be weighted less. This is the concept behind the proportional summation (PS) algorithm:(7)$$\theta_{mea}=\{B{\%}_{i}\times \left(\frac{\Delta \lambda_{i\_c}}{\Delta \lambda_{i\_c}+\ldots +\Delta \lambda_{n\_c}}\right)+\ldots +\{B{\%}_{n}\times \left(\frac{\Delta \lambda_{n\_c}}{\Delta \lambda_{i\_c}+\ldots +\Delta \lambda_{n\_c}}\right)\}\times \theta_{r-p}+\theta_{p}$$
If an FBG node does not receive the same strain from the joint, the summation is reduced with the strain in real-time. Equation (7) means that the proportion of $\theta_{r-p}$ is divided according to the peak shift in the present state compared to each FBG node corresponding to the joint. For example, if the MCP joint is 90° ($\theta_{r-p}=90{}^{\circ}$, $\theta_{p}=0{}^{\circ}$) during a closed fist, the reference angle is set as the highest peak shift among the corresponding nodes. Using this algorithm can reduce the errors implied by Equations (5) and (6).

However, differences could occur in nodes directly corresponding to the MCP joint between large and small hands. Figure 6 shows the peak shifts of nodes 5–9 corresponding to the MCP joint with large (117 mm) and small (77 mm) hands. For the small hand (77 mm), nodes 5 and 6 were directly strained by the MCP joint; for the large hand (117 mm), nodes 7 and 8 were directly strained by the MCP joint. Nodes 6 and 7 were indirectly or directly affected by the MCP joint with either hand size. In contrast, node 5 was at the center of the PIP and MCP joints for the large hand in a clenched fist, so it was not affected by any joint. In addition, if only the PIP joint moved, the peak shift may occur at node 5, which would increase the unwanted measured angle of the MCP joint.

To solve this problem, the bending percentage of the PIP joint should be used to reduce the influence of node 5. This is proportional summing/interference (PS/I) algorithm, which prevents interference by the PIP joint during the PS method and is expressed in Equation (9). In Equation (8), $C_{I\_\alpha }$ is the coefficient for the change in the bending percentage ($\Delta \lambda_{4\_c}/\Delta \lambda_{4\_m}$) of the PIP according to the hand size.
(8)$$C_{I\_\alpha }=\left(1-\alpha \frac{\Delta \lambda_{4\_c}}{\Delta \lambda_{4\_m}}\right)$$
(9)$$\theta_{MCP,\;IP}=\{B{\%}_{i}\times \left(\frac{C_{I\_\alpha }\times \Delta \lambda_{i\_c}}{C_{I\_\alpha }\times \Delta \lambda_{i\_c}+\ldots +\Delta \lambda_{n\_c}}\right)+\ldots +\{B{\%}_{n}\times \left(\frac{\Delta \lambda_{n\_c}}{C_{I\_\alpha }\times \Delta \lambda_{i\_c}+\ldots +\Delta \lambda_{n\_c}}\right)\}\times \theta_{r-p}+\theta_{p}$$

Nodes 5–8 (i=5, n=8) correspond to the MCP joint. The influence of the PIP joint angle on node 5 is calculated using Equation (9) based on Equation (7) with the coefficient from Equation (8). $\Delta \lambda_{4\_c}$ is the current peak shift of node 4, which is the bending percentage of the PIP joint. $\Delta \lambda_{4\_m}$ is the maximum peak shift of node 4 during the open palm and clenched fist in the reference process. Therefore, $\Delta \lambda_{4\_c}/\Delta \lambda_{4\_m}$ represents the bending percentage of the PIP joint. This term ranges from 0 to 1 and is subtracted from 1. Thus, a larger bending angle for the PIP joint reduces the effect of node 5.

The role of the bending percentage of the PIP joint was confirmed to depend on the hand size. For a small hand, the role of $\Delta \lambda_{4\_c}/\Delta \lambda_{4\_m}$ should be reduced because of the large influence of node 5. When the role of $\Delta \lambda_{4\_c}/\Delta \lambda_{4\_m}$ decreases, the MCP joint can be measured more accurately by using all of nodes 5–8. Conversely, for a large hand, the role of $\Delta \lambda_{4\_c}/\Delta \lambda_{4\_m}$ should be greater. Therefore, the angle of the MCP joint should be measured with only nodes 6–8, and node 5 should be excluded. By changing the value of α in Equation (8), the role of $\Delta \lambda_{4\_c}/\Delta \lambda_{4\_m}$ can be increased or decreased.

Different hand sizes between 77 and 117 mm were tested, and the results confirmed that a=0.00 was optimal for small hands (77 mm), a=0.65 was optimal for moderately sized hands (97 mm), and a=1.30 was optimal for large hands (117 mm). In particular, when α>1, the angle calculated from node 5 should be subtracted from the calculation for the MCP joint for a more accurate coefficient. The a values in the experiment were optimized by comparing the maximum peak shifts of nodes 6 and 7. As the hand size increased, the peak shift of node 6 decreased; as the hand size decreased, the peak shift of node 7 decreased with a clenched fist. For different hand sizes, the optimal values of α that most closely matched the experimental results were 1.70 (117 mm), 1.30 (107 mm), 0.59 (97 mm), 0.22 (87 mm), and 0.04 (77 mm). For the thumb, the optimal α values were 0.50 (78 mm) and 0.13 (68 mm). Thus, α for nodes 3 and 4 corresponding to the MCP joint of the thumb were set to 0.57 (78 mm) and 0.30 (68 mm) based on the relational formula $\Delta \lambda_{4\_m}/\left(\Delta \lambda_{3\_m}+\Delta \lambda_{4\_m}\right)$. The optimal a values obtained from the experiment led to the best accuracy for the MCP joint angle. However, they varied with hand size, so the input of α to each experiment was limited. A better approach would be to find a parameter with the most similar value to the optimal a and apply it through relational formulas for the peak shifts, such as $\Delta \lambda_{7\_m}/\Delta \lambda_{6\_m}$ for the index–little fingers and $\Delta \lambda_{4\_m}/\left(\Delta \lambda_{3\_m}+\Delta \lambda_{4\_m}\right)$ for the thumb.

The PS/I_α algorithm is the PS algorithm when α=0, and the PS/I algorithm is when α=1. Depending on the hand size, the PS or PS/I algorithm may be applicable. A suitable α can be automatically applied depending on the hand size, which will significantly reduce the error in angle measurement.

Figure 7 shows a flow chart for the process of selecting the appropriate algorithm using qualitative criteria. The PTS method uses only one node per joint, so the condition “Multiple FBG nodes are used to measure a joint.” has been added. The AREA method is used to calculate finger joint angles using a simple sum of multiple nodes without weighting certain nodes, so the condition “Sometimes, certain nodes are dislodged from the same curvature.” has been added. Finally, for PS and PS/I_α, it is important to ensure that a particular node is completely deviated from that curvature, so the condition “In some cases, certain nodes completely deviate from the influence of curvature.” has been added. This flow chart will help in selecting the appropriate algorithm for any situation when producing a wearable hand module described in this paper.

## 4. Joint Angle Measurement

### 4.1. Investigation with Different Hand Sizes

Figure 2 shows that the fixed point of the FBG strain sensor was the same from the index finger to little finger but differed for the thumb. Therefore, the 3D printed hand replica was tested by printing two different finger models: The index finger and thumb with length ranges of 77–117 and 68–78 mm, respectively. The FBG strain sensors used for the index-little fingers only differs in the number of nodes according to the finger length. Eleven adult males were selected to measure the lengths of their fingers, and the lengths of the index-little fingers had a range of 105 ± 11.4 mm while the thumb had a range of 75 ± 4.5 mm as shown in Table 2. Therefore, if the quantitative evaluation considered these ranges, the results should be applicable to all normal adult male hands. Some studies also included the length of the palm, but this was excluded from the present study because the data at each joint were read slightly differently as the sensors moved [22].

### 4.2. 3D Printed Hand Replica and Measurement Process

Reference angles were obtained using a goniometer for comparison with the measured angles obtained by the sensor data and algorithms. However, measuring angles of an actual human hand can introduce many errors. For example, the finger joint angle measured by goniometer deviated by 5–10° depending on whether the fist was clenched tightly or loosely. For a more accurate quantitative assessment, the wearable hand module was evaluated according to the finger joint angles of a 3D printed hand replica, which is shown in Figure 8.

The 3D printed hand replica was printed using VeroWhite Plus. The MCP, PIP, and DIP joint angles were maintained using angle fixation pins. The rotation pin was inserted into the center of rotation of each joint, and two joints were rotated in a concentric circle. Then, the angle fixation pin was inserted to secure a certain joint angle so that the sensor value of the hand module would be fixed. The joint angle was fixed at 0°, 20°, 40°, 60°, 80°, and 90°. The replica had pins to hold the angle, but there was some amount of play because of the resolution (30 µm) of the 3D printer. Since the hand module contained spring components, the MCP joint angle deviated by 10–15° during the mounting and removal processes. Therefore, the quantitative assessment would be more accurate if the hand module was equipped and the sensor values were logged while the actual angle was measured with the goniometer at the same time. Therefore, reference bars were designed and printed on the side of each joint phalanx so that the sensor data and goniometer measurements could be recorded simultaneously.

### 4.3. Quantitative Measurement with the Hand Replica

A wearable hand module was mounted on a hand replica printed in different sizes. The angles of the MCP, PIP, and DIP joints were measured using different algorithms (i.e., PTS, AREA, PS, PS/I, and PS/I_α). Figure 9 shows the measured angle errors of the finger joints at 0°, 20°, 40°, 60°, 80°, and 90° for different finger sizes: 77–117 mm for the index-little fingers and 68–78 mm for the thumb. All tests were performed five times at room temperature, and a statistical analysis was performed on the angle errors of the samples under various conditions.

Figure 9a shows that the DIP joint had the smallest mean angle error of 0.26° with the PTS algorithm, but the standard deviation was relatively large at ±3.43°. When an algorithm using multiple nodes such as AREA was used instead of just one node as PS, the deviation of the angle error became more stable. For the DIP joint, the optimal choice for the various hand sizes was confirmed to be the PS algorithm; the angle error was 0.82 ± 2.71° with a minimum error of 3.53° and maximum error of 1.89°. Nodes 1 and 2 corresponded to the DIP joint, even though node 1 was positioned a little bit away from the joint. For the PIP joint, the PTS algorithm had the largest error angle and standard deviation at 0.77 ± 4.57°, but a smaller angle error was observed when two or more nodes were used (such as with AREA or PS). With the PS algorithm, the angle error of the PIP joint under all finger length conditions was 0.65 ± 2.50° with a minimum angle error of 3.15° and maximum angle error of 1.85°.

For the MCP joint shown in Figure 9b, the PTS method showed a very large mean angle error and deviation of −1.17 ± 9.52°. This larger mean angle error decreased with AREA and PS but failed to meet the allowable angle error (which is larger than 5.0°, as mentioned in the introduction) for various hand sizes. With AREA, a small hand (77 mm) resulted in a large angle error of −6.23 ± 14.71°. With PS, a large hand (107 mm) resulted in a large angle error of −3.30 ± 3.10°. As discussed in Section 3.2, node 5 played a very important role; when the value from node 5 was affected by the degree of bending in the PIP joint (i.e., the PS/I_α algorithm was used), the angle error of the MCP joint was dramatically reduced. With PS/I_α, the MCP joints had an angle error of −0.21 ± 2.32° with a minimum angle error of 2.5° and maximum error of 2.1°.

For the DIP and PIP joints, the measured angles did not represent changes as significant as that of the MCP joint because they were fixed at the fingertip. For the MCP joint, the algorithm needed to be adjusted because the positions of the corresponding FBG nodes varied. Changing from PTS to PS/I_α reduced the maximum angle error of 8.19°. The behavior of the MCP joint tended to match the initial concept addressed in Figure 5 and Figure 6. Using the appropriate relational formula for FBG nodes rather than a single FBG node helped determine the correct angle. In particular, α had a very large influence; Figure 9c,d graphs the decrease in the angle error and the best and used a values for measuring the joint angle with PS. The PS/I_α algorithm is the PS algorithm at α = 0 and PS/I algorithm at α = 1; Figure 9c shows that PS/I_α minimized the angle error through the application of a constant.

Figure 10 graphs the relationship between the measured and actual angles corresponding to the best algorithm for each joint (PS for the DIP and PIP joints and PS/I_α for the MCP joint) in the five experiments. The linearity showed a slight decrease in linearity with the smaller hand (77 mm, Figure 10c). However, the average linearity had $R^{2}=0.9982\pm 0.001$ with the larger hand and $R^{2}$ = 0.9986 ± 0.001 with the smaller hand. Comparing the linearity $R^{2}$ values for different algorithms produced interesting results. Figure 10d shows the $R^{2}$ value for the MCP joint between the actual and measured angles using different algorithms. For the DIP and PIP joints, the measured angle error was not significant; the linearity $R^{2}$ values did not differ significantly according to the type of algorithm with 0.9971 ± 0.001 and 0.9966 ± 0.002, respectively. For the MCP joint, the linearity $R^{2}$ was 0.9778 ± 0.026 and showed a large difference depending on the algorithm. However, similar to the angle error shown in Figure 9c, the PS/I_α algorithm showed very strong linearity with 0.9983 ± 0.001. Moreover, when the algorithm was varied from PTS to PS/I_α, linearity of angular data approached that of the raw data (0.99975 ± 0.0005, in Section 2.4) of Bare FBG strain sensor obtained from the bending test. Compared to the linearity of the Bare FBG strain sensor, this linearity degradation of PTS and AREA algorithm was similarly caused by whether or not a particular node was completely under the influence of curvature. Figure 7 allows us to select algorithms such as PTS, AREA, PS, PS/I_α by noting whether a particular node is completely under the influence of curvature or indirectly affected by curvature or completely deviated from curvature. For Bare FBG strain sensor data, when measuring linearity before and after 10,000 bends, the two nodes were completely under the influence of curvature. Therefore, considering the position of the nodes on the curvature was very important to ensure high linearity of the data. In the case of the PTS and AREA methods, which did not consider the position of the nodes on the curvature, it was thought that the degradation of linearity occurred because it did not respond to the continuous movement of nodes on curvature. This outcome is precisely in line with our algorithm concept mentioned in Section 3.2 and is considered to be one of the reasons for the low error angle. Thus, the improved linearity can be used as a good basis for judging the suitability of an algorithm, in addition to the angle error.

This tendency was found for both the thumb and index-little fingers. Since the thumb is different, the IP joint of the thumb behaved the same as the MCP joint of the index-little fingers. Meanwhile, the MCP joint of the thumb behaved similarly to the DIP and PIP joints of the index-little fingers. Figure 11 shows the angle error and linearity of the wearable hand module attached to the thumb.

For the thumb, the PS/I_α algorithm had the smallest angle error. For the MCP joint located near the fixed point, AREA had the smallest angle error. For the IP joint, an angle error of 0.74 ± 2.84° was observed for the large thumb (78 mm), and an angle error of −0.35 ± 1.23° was observed for the small thumb (68 mm) with PS/I_α. The maximum angle error was 3.58°, and the minimum angle error was 0.88°. For the MCP joint, Figure 11b shows angle errors of 2.06 ± 2.53° and −0.05 ± 1.79° with thumb lengths of 78 and 68 mm, respectively. The maximum angle error was 4.59°, and the minimum angle error was 0.47°. Figure 11c,d shows no significant difference between the actual and measured angles. Unlike the index-little fingers, longer thumbs resulted in greater deviation from the target angle line. For the index-little fingers, a shorter finger reduced the distance between the PIP and MCP joints, and the role of node 5 constantly changed. For the thumb, however, the fixed point was on the opposite side, so the role of each joint changed, and the tendency was also reversed.

Thus, the algorithm required for each joint differed depending on the fixed point. A joint closer to the fixed point meant that relatively simple algorithms such as PS and AREA were more accurate. For joints further away from the fixed point, the PS/I_α algorithm was required to accommodate the changes in the roles of intermediate nodes.

### 4.4. Rendering of Fingers of Different Users in Real Time

The MFC-based program was used to observe the wearable hand module when worn on different hand sizes and track the finger angles in real time. In particular, the PIP joint bending without the MCP joint or the MCP joint bending without the PIP was properly tracked when the fist was clenched. For the MCP joint, the algorithms could properly track the finger joint angle because the locations of the FBG nodes corresponding to the MCP joint changed in real time when movements other than the clenching motion was used. Figure 12 shows that users with relatively large and small hands followed the virtual hand’s finger angles for various hand gestures. The results confirmed that the wearable hand module can be used to track finger joint angles in real time with high accuracy for hand gestures, not just clenching the fist.

## 5. Conclusions

Wearable hand modules using FBG strain sensors were fabricated and evaluated for their performance at accurately measuring the angles of finger joints in real time. Algorithms were developed for converting data from sensors that change position according to hand movements and sizes into joint angles, unlike conventional sensors that are fixed. PTS, AREA, PS, PS/I, and PS/I_α were used to measure the angles of the DIP, PIP, and MCP joints; PS was judged to be most suitable for the DIP and PIP joints, and PS/I_α was judged to be most suitable for the MCP joint. The maximum and minimum angle errors of the index-little fingers were 3.53 and 1.89°, respectively, for the DIP joint; 3.15 and 1.85°, respectively, for the PIP joint; and 2.53 and 2.11°, respectively, for the MCP joint. For the thumb, the maximum and minimum angle errors were 2.56 and 2.02°, respectively, for the IP joint and 3.43 and 1.41°, respectively, for the MCP joint. The MCP joint of the thumb had the largest maximum angle error of 1.01 ± 2.42°; the rest of the joints showed smaller angle errors. The average angle error of the wearable hand module for all joints, sizes, and conditions was 0.47 ± 2.51°, and MAE was 1.63 ± 1.97°. A rendering program confirmed that the wearable hand module can be fitted to users with various hand sizes and provides accurate tracking even when only the PIP or MCP joint is moved.

The results confirmed that the angles of all finger joints (MCP, PIP, DIP) could be measured with high accuracy even for different hand sizes, and the measurement accuracy differed depending on the algorithm. In addition to flexion/extension, future work should consider exact measurements of the abduction/adduction and hyperextension movements. The gap between FBG nodes should be further reduced, and corresponding algorithms should be developed to accommodate the hand sizes of adult females and adolescents.

## Figures and Tables

**Figure 1 sensors-20-01921-f001:**
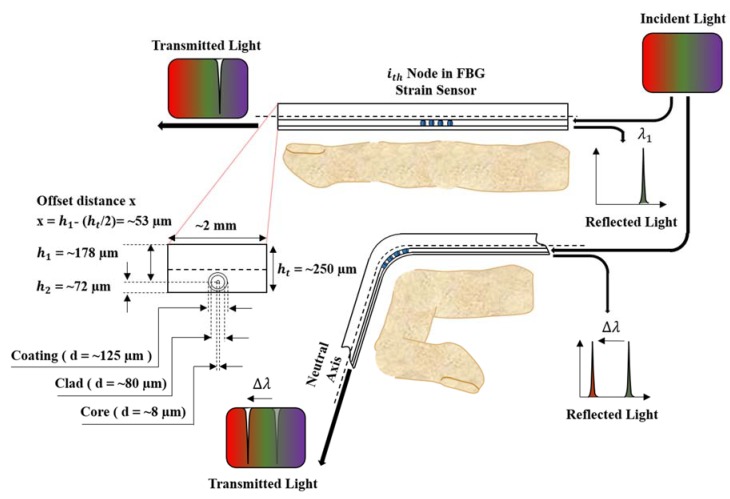
Principle of the fiber Bragg grating (FBG) strain sensors used to measure the finger joint angles.

**Figure 2 sensors-20-01921-f002:**
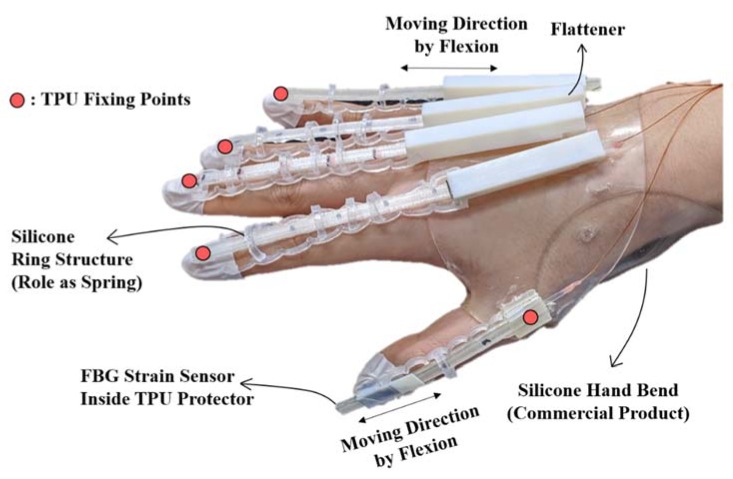
Wearable hand module and its components.

**Figure 3 sensors-20-01921-f003:**
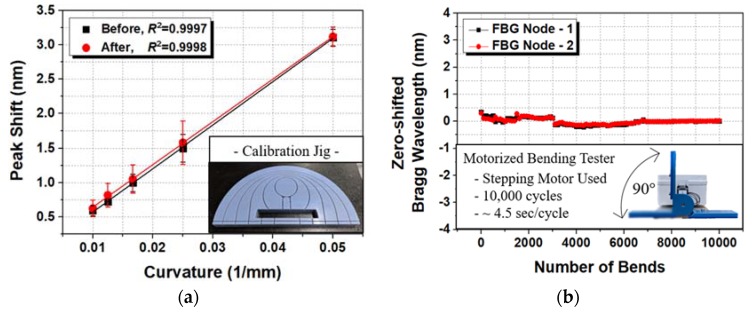
Evaluation of the FBG strain sensor: (**a**) Peak shift change by curvature and (**b**) zero-shifted Bragg wavelength during the high-cycle durability test.

**Figure 4 sensors-20-01921-f004:**
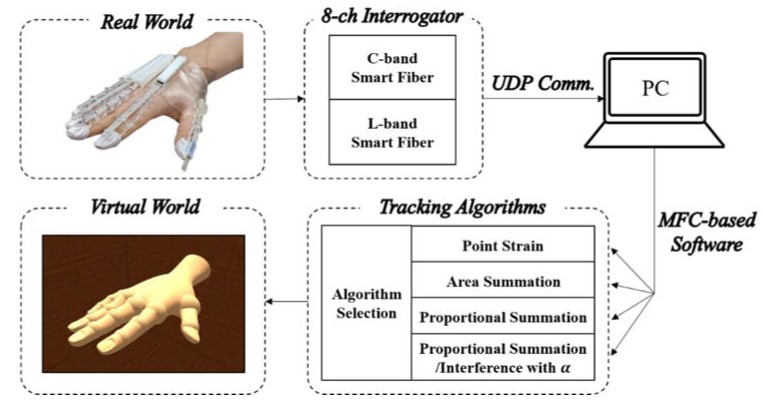
Schematic diagram of the FBG interrogation system for real-time tracking of the finger joint angle.

**Figure 5 sensors-20-01921-f005:**
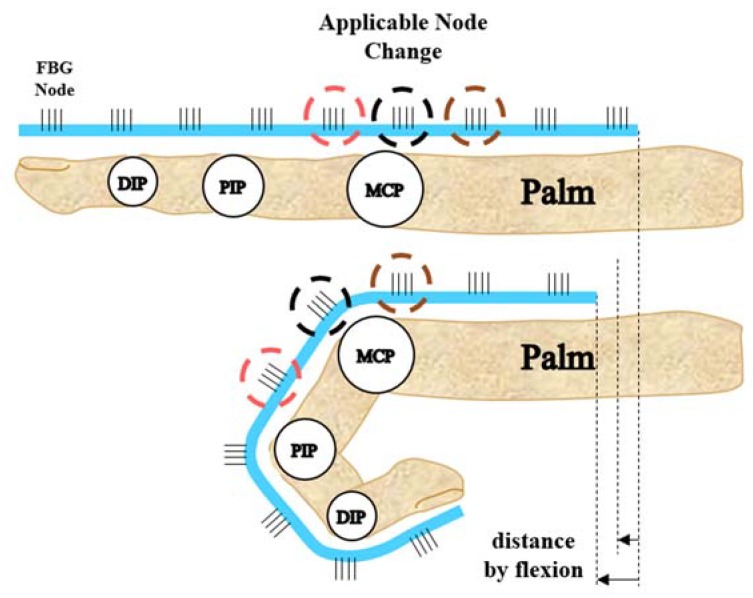
Positional changes of FBG nodes in the FBG strain sensor measuring the metacarpophalangeal (MCP) joint angle.

**Figure 6 sensors-20-01921-f006:**
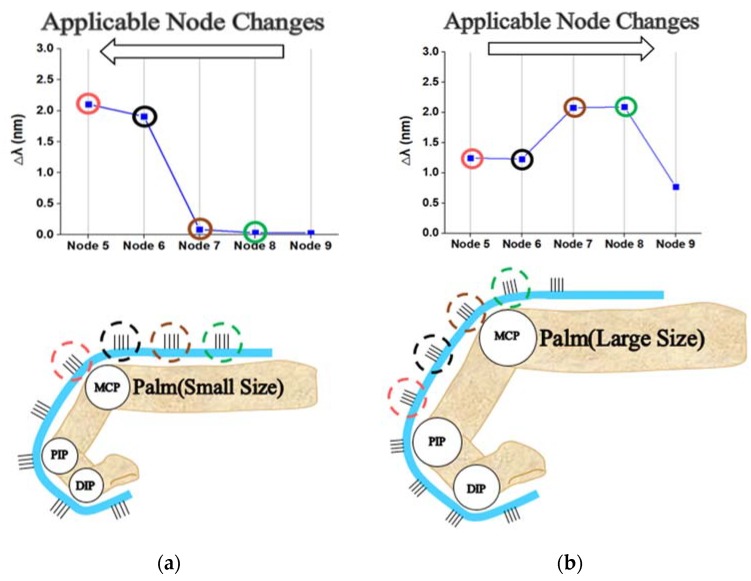
Peak shift changes of five nodes applicable to the MCP joint with different hand sizes: (**a**) Smaller (77 mm) and (**b**) larger (117 mm).

**Figure 7 sensors-20-01921-f007:**
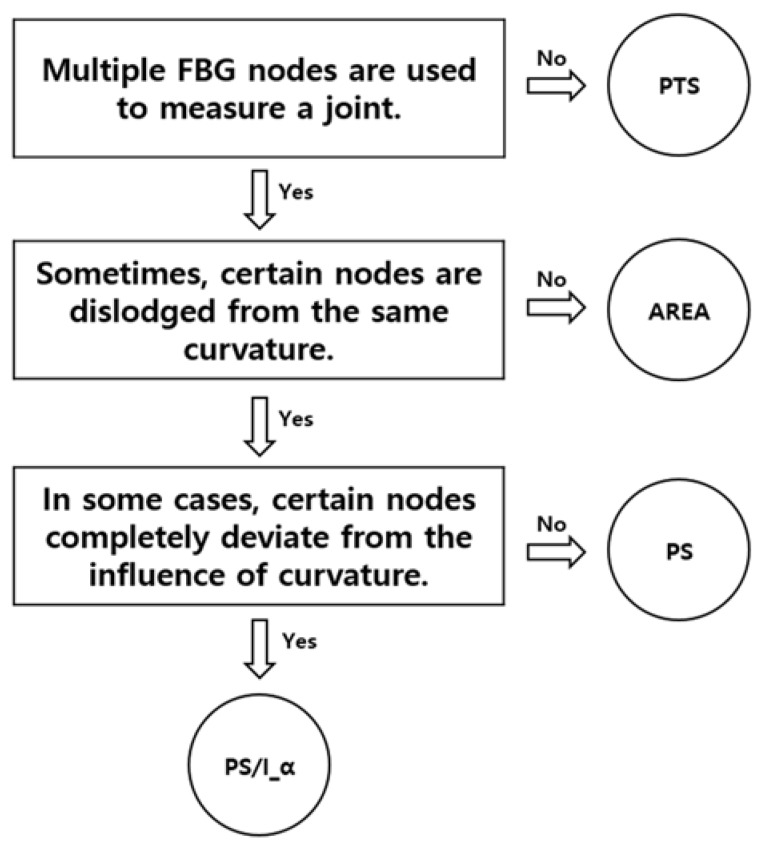
Flow chart diagram for selecting algorithms for measuring finger joint angles with FBG strain sensors.

**Figure 8 sensors-20-01921-f008:**
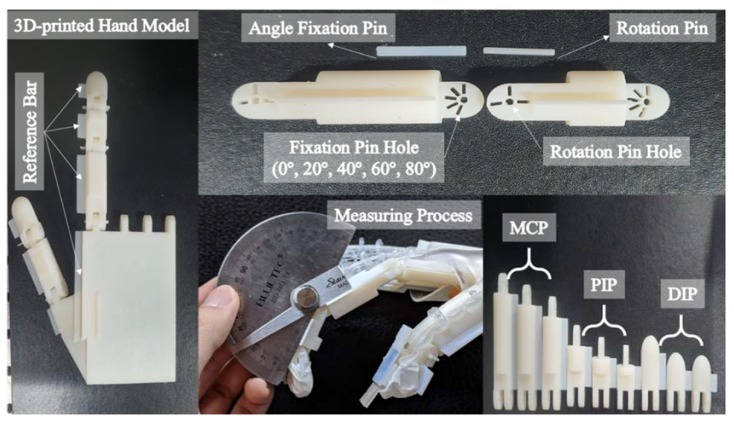
3D printed hand model used for quantitative evaluation of the wearable hand module.

**Figure 9 sensors-20-01921-f009:**
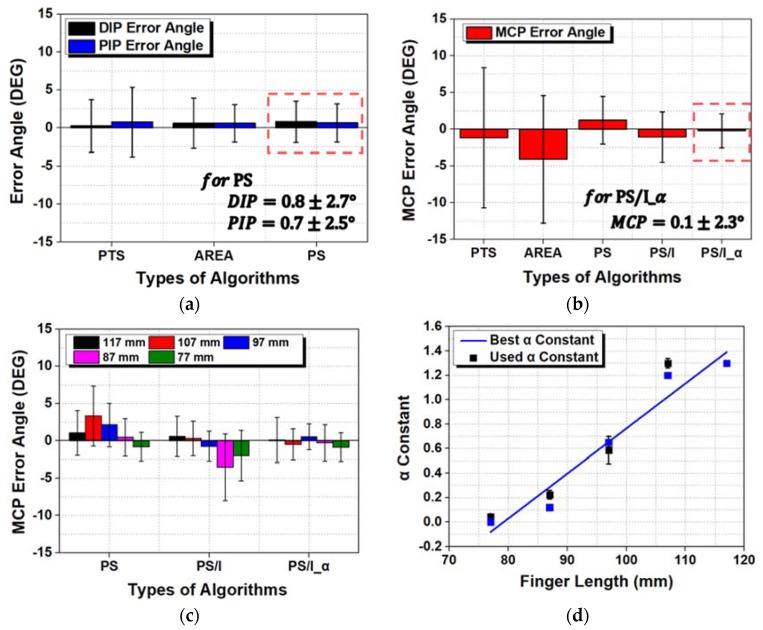
Joint angle errors of the wearable hand module according to different lengths of the index-little fingers: (**a**) Distal interphalangeal (DIP) and proximal interphalangeal (PIP) joint angle errors, (**b**) MCP joint angle error, and (**c**) MCP joint angle error according to the lengths of the index-little fingers, and (**d**) best and used α constant values for accurate measurement of the MCP joint.

**Figure 10 sensors-20-01921-f010:**
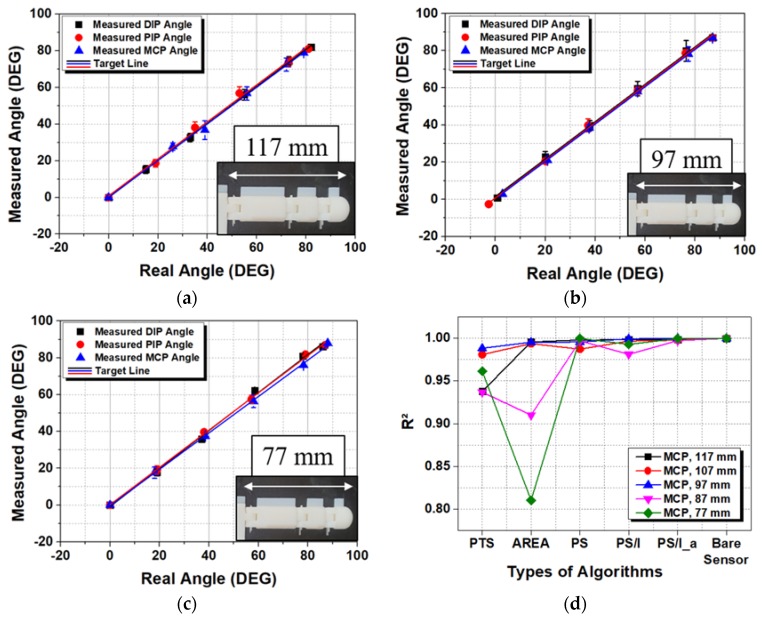
Measured angles versus real angles (measured by a goniometer) with respect to the length of the index finger: (**a**) 117, (**b**) 97, and (**c**) 77 mm. (**d**) Linearity of the measured MCP joint angle according to the algorithm and finger length.

**Figure 11 sensors-20-01921-f011:**
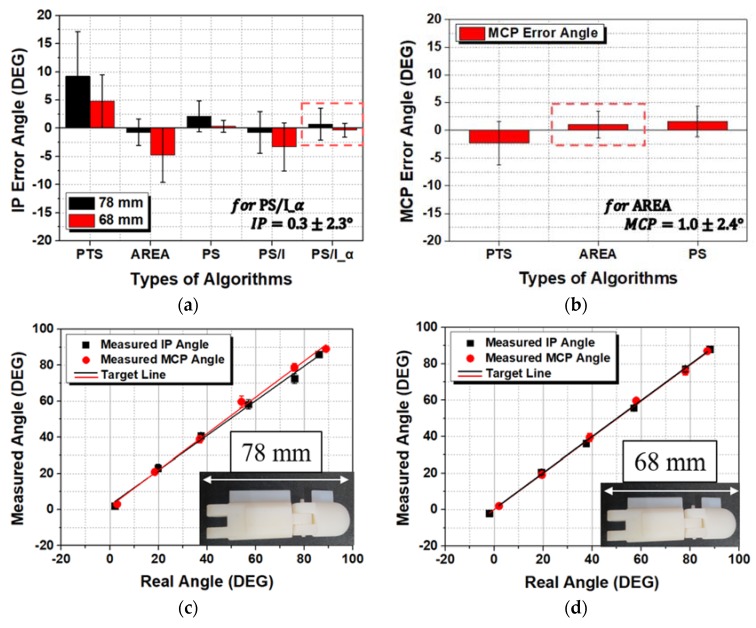
Measured angles versus real angles (measured by a goniometer) with respect to the size of the thumb: Angle errors of the (**a**) interphalangeal (IP) joint and (**b**) MCP joint; linearity at (**c**) 78 and (**d**) 68 mm according to the algorithm and type of joint.

**Figure 12 sensors-20-01921-f012:**
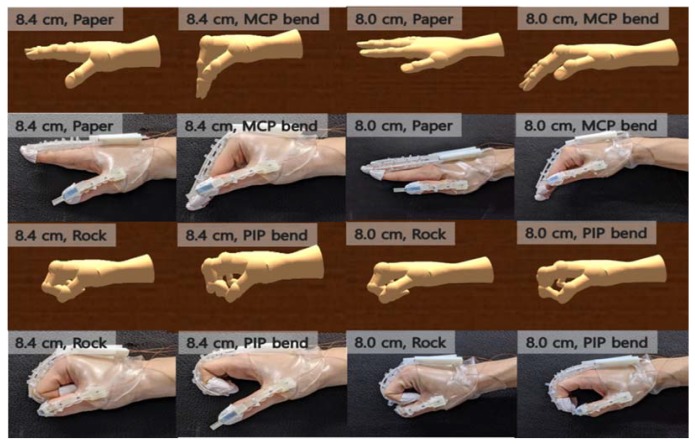
Comparison between real and rendered finger angle images.

**Table 1 sensors-20-01921-t001:** Specifications of the FBG sensors comprising the wearable hand module.

Finger Type	Sensor Length	Node Length (Distance)	Wavelength Range (Period)	Number of Nodes
Index Finger	155 mm	3 mm (19 mm)	1532.0~1562.4 nm (3.8 nm)	9
Middle Finger	155 mm		1532.0~1562.4 nm (3.8 nm)	9
Ring Finger	155 mm		1532.0~1562.4 nm (3.8 nm)	9
Little Finger	125 mm		1576.0~1600.0 nm (4.0 nm)	7
Thumb	85 mm		1584.0~1600.0 nm (4.0 nm)	5

**Table 2 sensors-20-01921-t002:** Different hand sizes of adult males investigated in this study.

Subject	Investigated Value
Number of People	11
Age Range	26~48
Index~Little Finger Length	105 ± 11.4 mm
Thumb Length	75 ± 4.5 mm

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
