# Peer review of "Wearable Hand Module and Real-Time Tracking Algorithms for Measuring Finger Joint Angles of Different Hand Sizes with High Accuracy Using FBG Strain Sensor"

_sensors, 2020, doi:10.3390/s20071921_

Round 1

Reviewer 1 Report

The manuscript presents hand modules using FBG strain sensors and the algorithms for measuring finger joint angles. The topic of study is very interesting and results justify publication. I only recommend to the authors to rewrite line 358. It seems the word "was" must be erased.

Author Response

The manuscript presents hand modules using FBG strain sensors and the algorithms for measuring finger joint angles. The topic of study is very interesting and results justify publication. I only recommend to the authors to rewrite line 358. It seems the word "was" must be erased.

  • We are very pleased to hear that you found the topic of our study very interesting. We rewrote line 358 from “… FBG strain sensor was is the same …” to “… FBG strain sensor was the same …”. This change has been marked with red color font in line 385 of the revised manuscript. We have also taken care of a few other miscellaneous grammatical errors and typos that we found in our review. In addition, very minor revisions, such as appropriate replacement words, proper prepositions, upper/lower-cases, and articles, were applied to the revision paper. Thank you for allowing us the opportunity to revise and improve the quality of our paper.
  • (please see the attachment files)

Reviewer 2 Report

The authors introduced a FBG sensor integrated wearable hand module which is applicable for various hand sizes. In general, it is a interesting application. However, there are still some concerns need to be addressed.

  1. In Sec. 2, line 141, the authors state that the FBG should be able to measure both axial strain and bending strain. However, by reading through the paper, I could not see the necessity of axial strain measurement. Can authors give more explaniations on this issue?
  2. I miss the definition of ht in Eq. (3). And is it necessary to introduce the variable x ? It is not being effectively used.
  3. In line 160, it says "a height of approximately 250 μm". But I can see in Fig. 1 that the height is ~ 200 μm. Which is the correct one?
  4. In Table 1, FBGs for index, middle and ring fingers are in C-bands, whereas the ones for little finger and thumb are in L-band. Can authors provide the reason? I believe it is one critical aspect in this approach.
  5.  In line 191, " ... bending cycles were performed on two nodes...". Which of the nodes were tested?
  6. From line 196 to 198, "... linearity of the system results may be significantly degraded". How is it degraded? I suppose it is critical to show the degradation as comparison, in order to show the effectiveness of the proposed algorithms.
  7. I suggest authors adding flowchart diagrams to help explaining different algorithms in Sec. 3.2.

Author Response

The authors introduced a FBG sensor integrated wearable hand module which is applicable for various hand sizes. In general, it is a interesting application. However, there are still some concerns need to be addressed.

  • We appreciate your helpful and detailed comments. We have revised our manuscript according to your suggestions, as listed below, and all changes are marked in red color in the revised manuscript.

1. In Sec. 2, line 141, the authors state that the FBG should be able to measure both axial strain and bending strain. However, by reading through the paper, I could not see the necessity of axial strain measurement. Can authors give more explaniations on this issue?

  • Even if the FBG strain sensor can measure both axial and bending strain, we can only see the bending strain data from FBG strain sensors. This is because of the attachment method of the FBG strain sensor in the TPU guide. In Figure 2 of the manuscript, the TPU guides (which protects the FBG strain sensors) are fixed at the nail points (with respect to the index–little finger). But the fixing point of the FBG strain sensor inside the TPU guide is in the totally opposite direction. The figure below will help you better understand the internal structure of the TPU guide.
  • Figure a (Please see the attachment)
  • As you can see in this figure, by the difference of fixing points of FBG strain sensors, We can only selectively receive strain by bending. If not, we must measure the axial strain and distinguish between axial strain and bending strain somehow like you concerned. We found that this has not been adequately addressed in the paper, so we have added the following text explaining how we attached the FBG strain sensors inside the TPU (line 182 of the revised manuscript).

2. I miss the definition of ht in Eq. (3). And is it necessary to introduce the variable x ? It is not being effectively used.

  • We included a definition of ht in line 154 line of revised manuscript as shown below. (Please see the attachment)
  • Regarding the role of offset distance x, when manufacturing the FBG strain sensors, offset distance x was a very important factor for defining sensitivity of the sensors. This is because, as mentioned in the manuscript, we can only control the offset distance of the FBG strain sensors when we fabricate the sensors. If the offset distance increases, the sensor becomes too sensitive and vice versa. If the sensor is too sensitive, there will be a collision between adjacent peaks of the output signal (you can see this phenomenon below in Figure b).
  • Figure b (Please see the attachment)
  • If the sensor is not sensitive, hand tremor will be observed in the 3D graphic hand model due to the interference of noise. This is why the offset distance x is important. In order to clarify these points, lines 157 to 163 were revised as shown below. Also, the exact offset distance was indicated Figure 1 as shown below.

3. In line 160, it says "a height of approximately 250 μm". But I can see in Fig. 1 that the height is ~ 200 μm. Which is the correct one?

  • The correct height is 250 μm. As mentioned in question 2’s answer, we made multiple attempts to find out the most appropriate FBG strain sensor for our wearable hand module for measuring the finger motion of people with different hand sizes. In the process, FBG strain sensors of several thicknesses were fabricated. The 200 μm thickness sensor was one of them. The reason why we didn’t mention this fabricating process is because we have already published a paper that discussed the fabrication and evaluation of the FBG strain sensors.

4. In Table 1, FBGs for index, middle and ring fingers are in C-bands, whereas the ones for little finger and thumb are in L-band. Can authors provide the reason? I believe it is one critical aspect in this approach.

  • To measure finger bending, we need 5 to 9 nodes as shown in Table 1. This is the whole wavelength area of one channel in the C-band area, and the interrogators used in our experiments were both 4ch C-band and 4ch L-band interrogator. The index, middle, and ring fingers must have nine nodes, which is a wavelength area that uses one whole channel of the C-band. Therefore, three of the four channels are already used for just these three fingers. For this reason, the thumb and pinky finger was measured using an L-band interrogator. In fact, if the specifications of the FBG sensor are all the same and only the wavelength area is C-band, then the thumb and the little finger can be measured in C-band as well. The change in band area from C to L does not make any difference. Likewise, changing from L-band to C-band also does not change anything. This is because we use two interrogators manufactured by the same company and having the same model. We have included below the specifications of interrogator used in our work.
  • Figure c (Please see the attachment)

5. In line 191, " ... bending cycles were performed on two nodes...". Which of the nodes were tested?

  • For the bending test experiment, we produced and tested the FBG strain sensor using a new FBG sensor, not reusing the FBG sensor inside the wearable hand module. I believe that the data will be reliable enough because it has been measured in the same environment (such as interrogator, etc.) and conditions (such as node length, grating period, sensor length, thickness, etc.) as the FBG sensors actually used inside the wearable hand module. Wavelength of the FBG Nodes for the bending tests are corresponded to the 4, 5 nodes in index finger sensor inside wearable hand module. Nodes only differ from its wavelength. And it doesn’t change anything because same interrogators and same grating condition was employed to whole different nodes which have different wavelength.

6. From line 196 to 198, "... linearity of the system results may be significantly degraded". How is it degraded? I suppose it is critical to show the degradation as comparison, in order to show the effectiveness of the proposed algorithms.

  • In fact, the comparison between linearity of sensor raw data (Peak Shift) by bending and linearity of angular values (secondary data obtained through algorithms) is very important. However, it was difficult to directly compare the linearity of the raw data with the linearity of the secondary angle data. The problem was that the linearity of the data did not have a sufficient range for comparison. In Section 4.3, we stated that DIP and PIP joints linearity data were excluded because they had such a high linearity that it was difficult to compare with each other. For example, DIP joint had a linearity value of 0.9971±0.001 and PIP joint had a linearity value of 0.9966±0.002. Given the problem of degraded linearity by the joints rather than algorithms, we were concerned about the significance of the difference in the linearity values, which was only 0.0005. For MCP joints, the range of linearity, which was 0.8105 to 0.9998 based on the algorithms, was sufficient for comparison without much concern as shown in Figure 9(d). Therefore, in the case of MCP joints, we can compare the degradation as shown in Figure d. Through this graph, we can directly compare the degradation between linearity of Bare FBG strain sensor data and linearity of FBG strain sensor data inside the wearable hand module, which is a secondary data from the algorithms. These points were explained in lines 467 to 480 of the revised manuscript as shown below. (Please see the attachment)
  • Figure d (Please see the attachment)

7. I suggest authors adding flowchart diagrams to help explaining different algorithms in Sec. 3.2.

  • As suggested, we have added a flow chart diagram (as Figure 7) of the proposed algorithms presented in the paper. This would make it easier and clearer when selecting algorithms to measure the finger joint angles using FBG sensors, which is moving continuously. Additionally, we have added an explanation regarding this flow chart as shown below (lines 371 to 382 of the revised manuscript). (Please see the attachment)

Once again, we would like to thank you for making our paper focus clearer and improving its level. Your comments have increased the depth of our research. In addition to the above-mentioned revisions, very minor revisions, such as appropriate replacement words, proper prepositions, upper/lower-cases, and articles, were applied to the revision paper.

Reviewer 3 Report

The authors report a type of wearable hand module using 5 FBG strain sensors and algorithms with high accuracy which is applicable to different hand sizes of user. The design and assessment of their module is interesting and impressive to me. I suggest to accept this work for publishing in Sensors. However, the authors may need to polish their writing further to avoid grammar errors or typos.

Author Response

The authors report a type of wearable hand module using 5 FBG strain sensors and algorithms with high accuracy which is applicable to different hand sizes of user. The design and assessment of their module is interesting and impressive to me. I suggest to accept this work for publishing in Sensors. However, the authors may need to polish their writing further to avoid grammar errors or typos.

  • We are very pleased to hear that the design and assessment of our module was interesting and impressive to you. We thoroughly reviewed the manuscript and made corrections to grammatical errors and typos. In addition, very minor revisions, such as appropriate replacement words, proper prepositions, upper/lower-cases, and articles, were applied to the revision paper. Also, additional explanations were included as needed to improve the readability of our manuscript. We are sincerely thankful for allowing us to write a better paper.

Reviewer 4 Report

The authors present considerable results of using fibre Bragg gratings for tracking the fingers angular movements. The paper is very well organized and presented. A great deal of work has been done. The results and the experimental methods used in the work will be beneficial to the other researchers in this specific area. The reviewer think it can be published in the present form.

Two mistypings: line 52 – “globe,” line 358: “was is”.

It’s better to define “ht” in equation (3) although it can be deduced from (4).

Author Response

The authors present considerable results of using fibre Bragg gratings for tracking the fingers angular movements. The paper is very well organized and presented. A great deal of work has been done. The results and the experimental methods used in the work will be beneficial to the other researchers in this specific area. The reviewer think it can be published in the present form.

Two mistypings: line 52 – “globe,” line 358: “was is”.

It’s better to define “ht” in equation (3) although it can be deduced from (4).

  • We are very pleased to hear that you found our paper to be well organized and presented. We have corrected the two typos you pointed out and marked them in red font in the revised manuscript. Specifically:
    • In line 52, we have corrected “... directly to the globe ...” to “... directly to the glove ...”.
    • In line 385, we have corrected “… FBG strain sensor was is the same …” to “… FBG strain sensor was the same …”.
  • We agree with your opinion that it would be better if we explicitly define “ht” in the manuscript. We have done so in line 154 of the revised manuscript, as shown below.
  • Apart from this, we have thoroughly reviewed the manuscript and fixed a few other miscellaneous grammatical errors and typos as well. All such changes have been marked in red font in the revised manuscript. In addition, very minor revisions, such as appropriate replacement words, proper prepositions, upper/lower-cases, and articles, were applied to the revision paper. Thank you for allowing us the opportunity to revise and improve the quality of our paper.
  • (Please see the attachment)

Round 2

Reviewer 2 Report

Since my concerns in the pervious review reports have been addressed, I recommend the publication of the paper.